# COVID-19 vaccination coverage in deprived populations living in segregated colonies: A nationwide cross-sectional study in Hungary

**János Sándor**[1]*, **Ferenc Vincze**[1], **Maya Liza Shrikant**[1,2], **László Kőrösi**[3], **László Ulicska**[4], **Karolina Kósa**[5], **Róza Ádány**[1,6]

1 Department of Public Health and Epidemiology, Faculty of Medicine, University of Debrecen, Debrecen, Hungary, 2 Arizona State University, Tempe, Arizona, United States of America, 3 National Health Insurance Fund, Budapest, Hungary, 4 Deputy State Secretariat for Social Inclusion, Ministry of Interior, Budapest, Hungary, 5 Department of Behavioral Sciences, Faculty of Medicine, University of Debrecen, Debrecen, Hungary, 6 MTA-DE Public Health Research Group, Department of Public Health and Epidemiology, Faculty of Medicine, University of Debrecen, Debrecen, Hungary

* janos.sandor@med.unideb.hu

**Data Availability Statement:** The database of the investigation with the segregated colony and complementary area specific observed and

## Abstract

The segregated colonies (SCs) in Hungary are populated mainly but not exclusively by Roma. Their health care use is restricted in many respects. It has not been studied yet, whether fair COVID-19 vaccination coverage achieved in Hungary is accompanied with fair effectiveness in SCs. Using census data, the vaccination coverage in SCs and the complementary areas (CAs) in the same settlements of the country was determined. To describe the settlement level differences, the vaccination coverage (until June 30, 2021) in SCs were compared to those in CAs by age, sex, and eligibility for exemption certificate standardized measures. Aggregating settlement level data, the level of geographic discrimination in Hungary was also determined. According to nationwide aggregates, crude vaccination coverage was significantly lower in SCs (40.05%, 95% CI 39.87%-40.23%) than in CAs (65.42%, 95% CI 65.37%-65.46%). The relative standardized vaccination coverage was 0.643 (95% CI 0.639–0.647) in SCs. A total of 437 of the 938 investigated settlements showed significant local vaccination disparities. Hungarian citizens living in SCs, mainly of Roma ethnicity, are a distinct high-risk group. Special intervention adapted to SCs is needed to mitigate inequality in vaccination coverage and further consequences of the pandemic.

## Introduction

The Severe Acute Respiratory Syndrome Coronavirus-2 (SARS-CoV-2) pathogen brought on a serious pandemic, which has been forecasted for a long time [1–3]. The echo of the concerned is reflected in the dramatic health, economic and social loss caused by the first three waves of the COVID-19 pandemic affecting all countries of the world [4]. The profound inequalities with respect to infection rate [5], mortality rate [6–11], and vaccination coverage are [12] among the dominant characteristics of this pandemic.

expected number of completed vaccinations had been archived in the figshare repository (https://doi.org/10.6084/m9.figshare.17158121.v1) This file can be downloaded without any restriction. All the indicators reported in our manuscript can be produced from this dataset. The name of settlements have been replaced with serial numbers in the dataset, because the identification of a settlement is not allowed: the Hungarian law on statistical data reporting (2016. évi CLV. törvény a hivatalos statisztikáról; Act on Official Statistics 2016) and related decrees prohibit the report of statistical data with less than or equal to 5 number of observed cases, and there are many observed number of cases meeting this criterion in the dataset.

**Funding:** This study was carried out in the framework of the " Routine monitoring for the health status and health care use in the Hungarian segregated colonies" program (BM/6327-3/2021, FEIF/951/2021-ITM), supported by the Deputy State Secretariat for Social Inclusion, Ministry of Interior. (https://2010-2014.kormany.hu/en/ministry-of-interior) JS, FV, LK, KK, RÁ received grant from that program. The funder had no role in study design, data collection and analysis, decision to publish, or preparation of the manuscript.

**Competing interests:** The authors have declared that no competing interests exist.

When and where COVID-19 vaccines became available, the vaccination programs' target groups were determined by taking into consideration the susceptibility of individuals to lethal complications (e.g., elderly individuals, patients with chronic diseases) and to the personnel involved in COVID-19 patients' health care [13].

The distinct (i.e., socioeconomic status independent) risk factor role of race/ethnicity has been demonstrated by investigations based on person-level race/ethnicity classification [14–16]. Studies, both on migrants by their country of birth [17] and geographical analyses of aggregated data on racial/ethnic composition and vaccination coverage [18–23], confirmed this observation. This evidence supports the case for strict monitoring of racial/ethnic inequalities in vaccination uptake/coverage, which is confirmed by published reports on successful interventions in this regard [24, 25]. To date, there are published reports on successful interventions aimed at diminishing racial/ethnic disparities in vaccination coverage [26–29].

These studies suggest that vaccination programs should consider racial/ethnic minorities as special high-risk groups that require minority-adapted, culturally appropriate approaches (complementing the main programs designed for the majority) to diminish the consequences of the vaccine-preventable COVID-19 infections pandemic in the forthcoming wave(s) of the pandemic.

Central and Eastern European (CEE) countries are home to 10–12 million Roma (many of them in segregated colonies, SC) who are the largest ethnic minority in Europe. In spite of some positive European experiences on Roma specific monitoring [30], there are no socioeconomic status (SES) or ethnic minority-specific monitoring programs either for COVID-19 health impacts or COVID-19 vaccination effectiveness in CEE countries. Although it seems probable that the CEE Roma are similar to the Black populations in the US and UK as well as to the Native Americans in the US with respect to health care use, the lack of monitoring prevents even planning targeted pandemic intervention [31–33]. Taking into consideration disadvantaged SES and the health status of Roma in CEE countries with limited access to health care [34], the COVID-19 vaccination coverage among this people group needs to be evaluated.

In Hungary, the last census in 2011 classified 315,583 persons as Roma (3.18% of the total population) by self-declaration. This approach (due to the negative historical experiences of ethnic minorities in governmental registration) seriously underestimates the number of Roma people. A special survey used the active contribution of the local municipalities estimated their number to be 870,000 (8.75% of the Hungarian population) [35, 36].

The poorer health status of the Roma living in Hungary is well documented [37, 38]. Approximately one-quarter of them [39] with a critical health status live in SCs. To improve the effectiveness of interventions targeting this high-risk population, a governmental decree (314/2012) defines the SCs utilizing census data. It establishes the production of indicators by which the needs can be explored and interventions can be elaborated for SCs, as well as the effectiveness of implemented SC-specific programs can be assessed.

Roma comprised 94% of the inhabitants in SCs in 2005 [9]. Although the Roma characteristics of SCs have weakened due to social changes over the last two decades, there have been no new investigations on the proportion of Roma inhabitants in SCs.

The free-of-charge voluntary COVID-19 vaccination program has been organized by the government in Hungary and it started on 26 December 2020. By the time our dataset was generated on 30 June 2021, Hungarian vaccination coverage was at 58.89% among adults, the 2nd highest in Europe according to the European Centre for Disease Prevention and Control [40].

Our investigation utilized the special opportunity of the Hungarian health statistical system (e.g., its ability to produce SC-specific indicators), to describe COVID-19 vaccination coverage in Hungary's SCs in relation to the nonsegregated part of the same settlements to determine

whether the population living in SCs dominated by Roma should be a distinct target group for vaccination programs.

## Methods

### Setting

This cross-sectional investigation encompassed the whole country. All COVID-19 vaccinations registered before 30 June 2021 were used in this study. The government organized a free-of-charge vaccination program using the Oxford/AstraZeneca COVID-19 vaccine, Janssen Ad26.COV2.S COVID-19 vaccine, Moderna COVID-19 (mRNA-1273) vaccine, Pfizer-BioN-Tech COVID-19 mRNA vaccine, Sinopharm COVID-19 vaccine, and the Sputnik V vaccine. The National Health Insurance Fund (NHIF) registered all vaccinations.

### Mapping the segregated colonies

A governmental decree defines SCs as within settlement (within towns and within villages) clustering of residents 18–59 years old with not higher than primary level education and a lack of work-related income. The Hungarian Central Statistical Office determines the cluster or clusters as SCs and the complementary areas (CA) of the same settlements not belonging to any SC for all Hungarian settlements. Each Hungarian household is classified in this system as either an SC or a CA, in a mutually exclusive manner.

   With the addresses of the adults the NHIF can define populations living in certain SCs or CAs. Therefore, all Hungarian adults can be classified as inhabitants living in an SC or living in a CA.

### Data sources

Data were provided by the NHIF to which reporting COVID-19 vaccinations is compulsory. The NHIF updated their database day-by-day. In this study, vaccinations until 30 June 2021 were included.

   The NHIF characterized all patients by their age, sex and eligibility for an exemption certificate (deprived patients with chronic disease are supported by an exemption certificate which is released by the local municipality based on the recommendation of the patient's GP), and prepared the age group-, sex-, and eligibility for an exemption certificate-aggregated tables for each SC and CA based on the administration of the first and second COVID-19 vaccination doses.

   The NHIF provided the target population data in the same structure. The age- (5-year bands), sex-, and possession of exemption certificate-specific numbers of vaccinations and populations were provided for each SC and CA.

### Statistical analysis

People who have received both doses of the vaccine are defined as 'vaccinated' in this analysis. The crude vaccination coverage (cVC) was calculated for each SC and CA. The different vaccine types were not distinguished. Age, sex, and eligibility for exemption certificate-specific reference vaccination ratios for the whole population of Hungary were calculated and used in indirect standardization. Age, sex, and exemption certificate standardized vaccination coverage (sVC) was computed for each SC and CA.

   The settlement-specific sVC was evaluated by comparing to the local reference value of settlement-specific sVC in CAs. Relative standardized vaccination coverage (RsVC), the ratio of sVCs for the SC to sVCs for the CA of the same settlement, along with the corresponding 95%

confidence interval (95% CI), were calculated. (In the settlements where more than one SC was located, settlement-specific aggregated SC measures were computed by summarizing the observed number of vaccinations, populations, and the expected number of vaccinations.) The excess number of vaccinations was calculated as the difference between the observed and the expected number of vaccinations. The proportion of vaccinations attributable to segregation (attributable risk) was computed as the ratio of excess and observed number of vaccinations.

SPSS version 20 (IBM Corporation, New York, NY, USA) was used for the data analysis.

### Ethical permission

Data provided by the Hungarian NHIF were used in this study. All data processed in our secondary analyses were geographically aggregated. Because individual information was not used, ethical approval and written informed consent were not required according to the Hungarian legal framework. The protocol to produce segregation-specific indicators was approved by the Office of the Commissioner for Fundamental Rights (AJB-3147/2013), the general director of the NHIF (E0101/215-3/2014), and the Hungarian National Authority for Data Protection and Freedom of Information (NAIH/2015/826/7N). Additional information regarding the ethical, cultural, and scientific considerations specific to inclusivity in global research is included in the (S1 Checklist).

## Results

There are 2006 SCs in Hungary belonging to 938 settlements. A total of 4,987,661 persons (60.7% of the Hungarian adult population) lives in settlements with SCs, and 276,879 (5.6%) of them live in SCs. This proportion was higher in the 18-59-year-old age group (6.6%) than in the older age group (3.45%), reflecting the poor life expectancy in SCs. Younger adults were overrepresented, and elderly adults were underrepresented in the SC population. (Fig 1) The proportion of adults who were eligible for an exemption certificate was 6.43% (N = 17,814) in SCs and 2.39% (N = 112,700) in CAs, showing a higher level of deprivation in the SC population (p<0.001 by the chi square test).

The overall cVC was 64.0% (3,192,457 vaccinations). According to nationwide aggregates, cVC was significantly lower in SCs (40.05%, 95% CI 39.87%-40.23%) than in CAs (65.42%, 95% CI 65.37%-65.46%). The difference was much larger in the 18- to 59-year-old age group than in the older than 60 age group (Table 1).

The sVC proved to be significantly lower in SCs (0.653, 95% CI 0.650–0.657) than in CAs (1.017, 95% CI 1.016–1.018). The RsVC was 0.643 (95% CI 0.639–0.647), associated with 58,807.9 missing vaccinations in SCs. Living in a SC accounted for 53% of the missing vaccinations. The difference was mainly generated in the 18-59-year-old age group, but it was significant in the older than 60-year-old age group as well (Table 2).

The SC-specific RsVCs showed a wide distribution among adults (mean±SD: 0.771±0.251). The mean RsVC was lower for the 18-59-year-old age group (0.740±0.262) than for the older than 60-year-old adults (0.884±0.217). The majority of SCs had lower local reference sVCs compared to CAs (Fig 2).

Mapping the SCs, where local interventions were established by sVC were significantly lower than the sVCs in the CAs in the corresponding settlement; the spatial distribution proved to be highly uneven (Fig 3A–3C).

Altogether, the number of SCs with significantly lower sVCs was 437, with 208,494 inhabitants, of which 80,065 were vaccinated. The number of missing vaccinations was 47,494.5. Most of the missing vaccinations were connected to the 18-59-year-old age group. In the 60 + age group, there were much fewer missing vaccinations (Table 3).

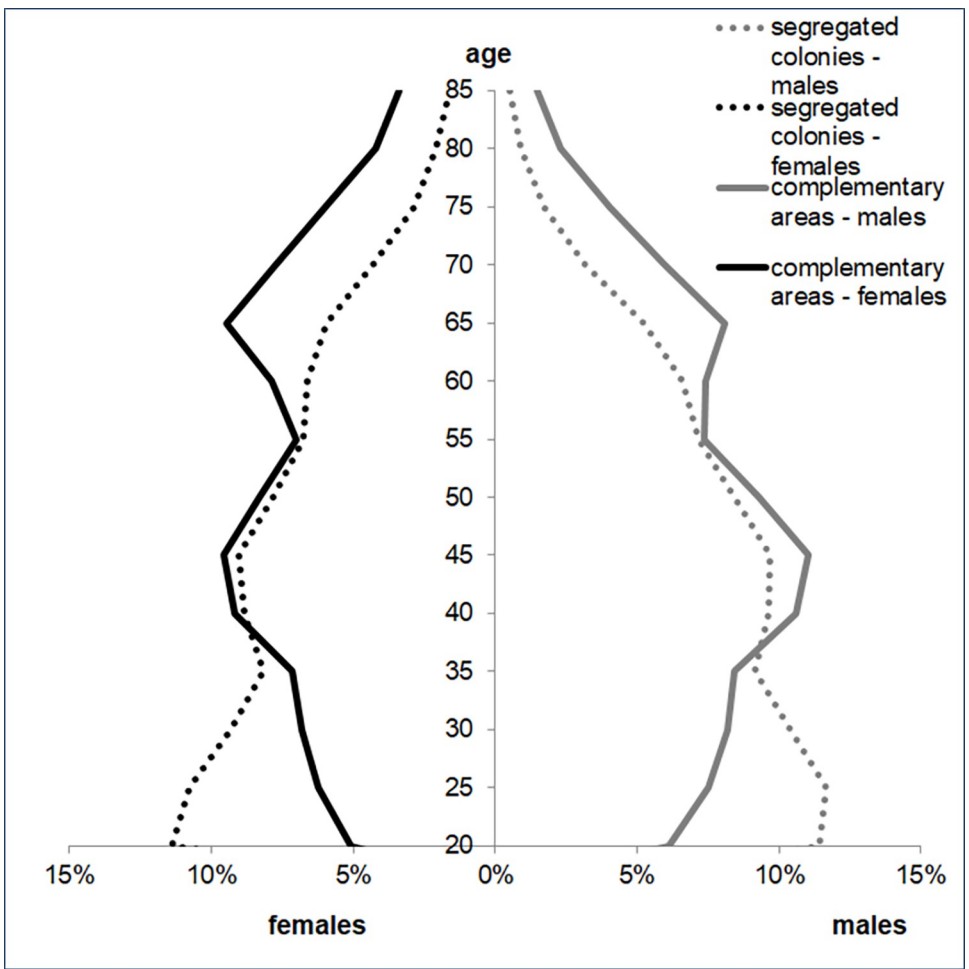

**Fig 1. Demographic structure of the populations living in segregated colonies and complementary areas of the settlements with segregated colonies in Hungary.**

## Discussion

### Main findings

Our study shows that COVID-19 vaccination coverage is significantly lower in SCs than in CAs according to both cVC (40.05% vs. 65.42%) and sVC (0.653 vs. 1.017). This demonstrates

**Table 1. Crude COVID-19 vaccination ratios among adults in segregated colonies and in the complementary part of the settlements with segregated colonies (Hungary, as 30 June 2021).**

| Age groups | Number of observed cases in the segregated colonies | Target group in the segregated colonies | Vaccination coverage in the segregated colonies* | Number of observed cases in the complementary areas | Target group in the complementary areas | Vaccination coverage in the complementary areas* |
|---|---|---|---|---|---|---|
| **18–59 years** | 75,567 | 219,156 | 34.48% [34.28%-34.68%] | 1,888,229 | 3,094,472 | 61.02% [60.97%-61.07%] |
| **60 + years** | 35,324 | 57,723 | 61.20% [60.80%-61.59%] | 1,193,337 | 1,616,310 | 73.83% [73.76%-73.90%] |
| **18 + years** | 110,891 | 276,879 | 40.05% [39.87%-40.23%] | 3,081,566 | 4,710,782 | 65.42% [65.37%-65.46%] |

*with 95% confidence interval.

**Table 2. Standardized COVID-19 vaccination coverage among adults in the segregated colonies and in the complementary part of the settlements with segregated colonies (Hungary, as 30 June 2021).**

| Age groups | Standardized vaccination coverage in segregated colonies* | Standardized vaccination coverage in complementary areas* | Relative standardized vaccination coverage in segregated colonies* | Number of excess cases in segregated colonies | Attributable risk in segregated colonies |
|---|---|---|---|---|---|
| 18–59 years | 0.593 [0.589–0.597] | 1.026 [1.024–1.027] | 0.578 [0.574–0.582] | -51,882.7 | -68.7% |
| 60 + years | 0.836 [0.827–0.845] | 1.003 [1.001–1.005] | 0.833 [0.825–0.842] | -6,925.2 | -19.6% |
| 18 + years | 0.653 [0.650–0.657] | 1.017 [1.016–1.018] | 0.643 [0.639–0.647] | -58,807.9 | -53.0% |

*with 95% confidence interval.

that the Hungarian COVID-19 vaccination program did not avoid geographic inequality, which has a strong ethnic dimension. This observation confirms the reported experiences from other countries about the inequality-generating characteristics of COVID-19 vaccination programs [17–23].

The cVC was much higher and vaccination inequality was much less pronounced among elderly individuals older than 60 years, compared to adults in the 18–59 age range. Organizational efforts were much more intensive and started earlier among elderly adults than among younger adults. The aim of public health authorities is not only to reduce of the epidemic's intensity (reduction of susceptible population) but also to prevent serious complications, especially the infection of elderly individuals who were at higher risk. Personal motivation was also larger among elderly individuals for the same reasons.

By the time of our investigation, a total of 165,988 adults (9.24% of the sensitive population) were from the SCs and 1,629,216 from CAs. If the vaccination coverage in SCs were equal to that in CAs, then there would be only 95,758 susceptible inhabitants in SCs. It would be reflected in 3.91% decrease of the number of susceptible adults in the studied population, and in the 5.55% of the susceptible inhabitants would be from SCs. Consequently, although, the

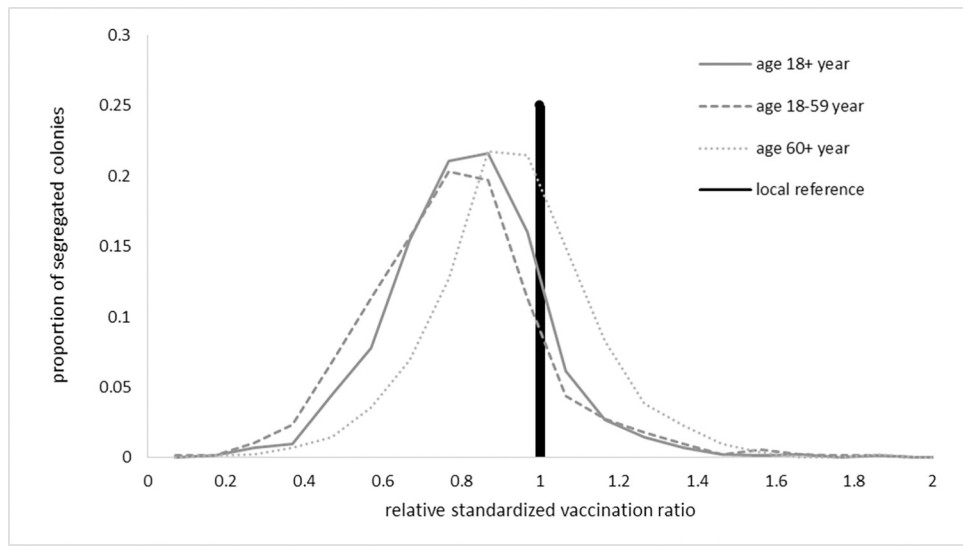

**Fig 2. Vaccination coverage among adults in segregated colonies compared to the complementary area of the settlements with segregated colonies in Hungary.**

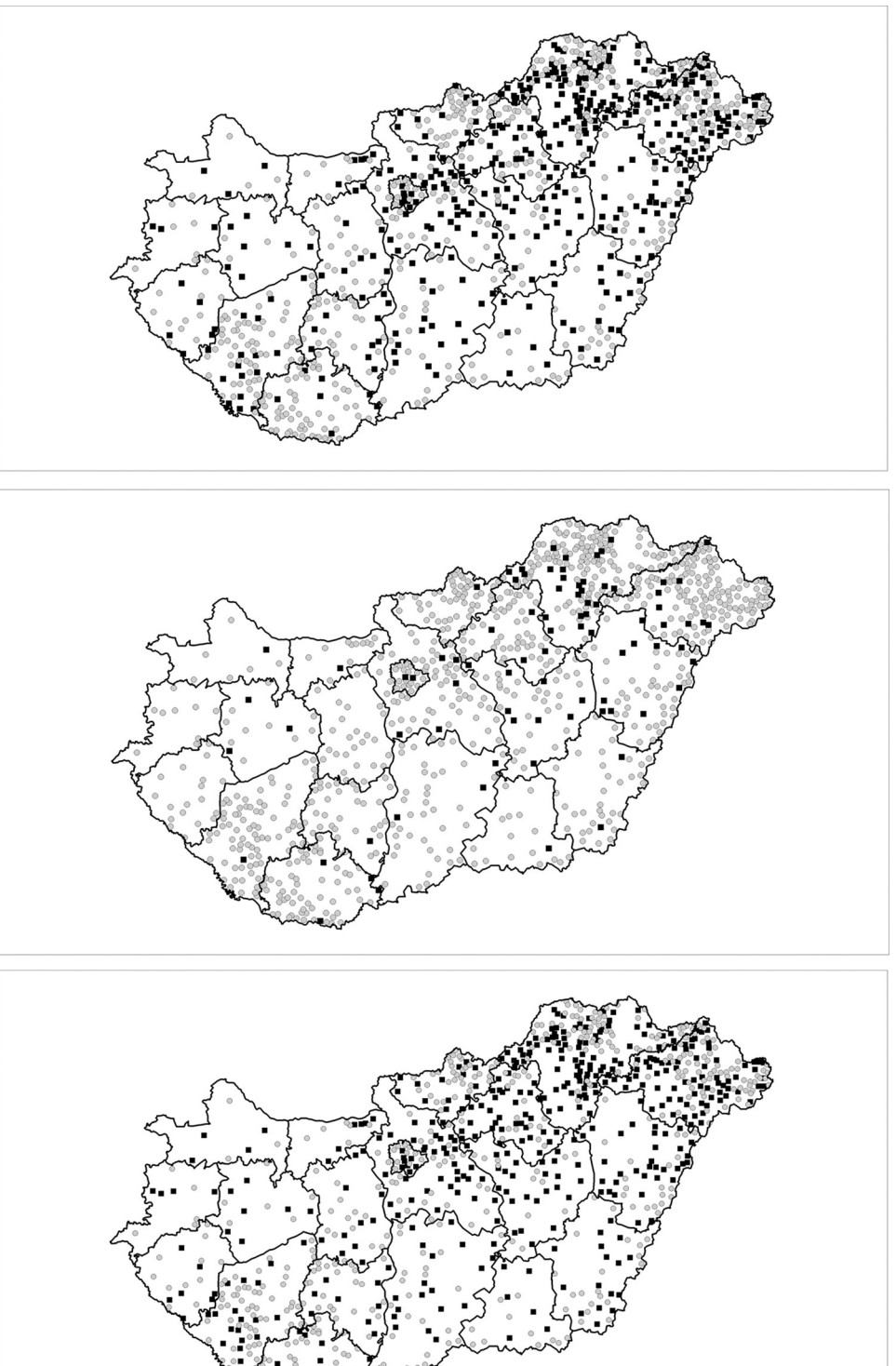

**Fig 3. a.** Settlements with local intervention requirements in Hungarian counties established by low relative standardized vaccination coverage among 18- to 59-year-old adults. (black square: standardized vaccination coverage in segregated colonies less than the complementary area's reference; white circle: standardized vaccination coverage in segregated colonies not deviating from the complementary area's reference). **b.** Settlements with local intervention requirements in Hungarian counties established by low relative standardized vaccination coverage among 60+ adults.

(black square: standardized vaccination coverage in segregated colonies less than the complementary area's reference; white circle: standardized vaccination coverage in segregated colonies not deviating from the complementary area's reference). **c.** Settlements with local intervention requirements in Hungarian counties established by low relative standardized vaccination coverage among 18+ adults. (black square: standardized vaccination coverage in segregated colonies less than the complementary area's reference; white circle: standardized vaccination coverage in segregated colonies not deviating from the complementary area's reference).

pandemic related future health loss could be prevented basically by the elevation of the vaccination in the CAs, a not negligible risk reduction could be achieved by the enforced/adapted vaccination project in the SCs.

Our investigation described the huge variability of settlement-specific RsVC. SC-related under-vaccination was generated by 437 SCs, where the number of adult inhabitants was 208,494 and the number of missing vaccinations was 47,494.5. The capacities required to make up the missing vaccinations seem to be small, given that in Hungary, 4,746,822 adults have been vaccinated at the time of our study.

It is probable that the vaccination inequality demonstrated by our study will increase the intensity of the fourth wave of the pandemic and result in health losses in Hungary, as it was observed in the second wave as area deprivation related COVID-19 mortality inequality [41].

## Strengths and limitations

This investigation covered the entire country, which prevented selection bias. Because vaccination registration is compulsory in Hungary, misclassification of the vaccination status was also avoided.

The unique feature of the Hungarian health statistical system is that the segregated colonies populated mainly by Roma are defined in the census data, and health indicators can be computed for these geographical units. Since, the registration of ethnicity in Hungarian health care administration is prohibited as it is in other European countries as well; the main strength of our study design was that it could avoid the violation of personal rights by utilizing SC related opportunity.

The living place of adults could be misclassified if the registered residential address was in a SC but the real residential address was in a CA (or vice versa). The extent of this residential place misclassification is not known. Therefore, the potential dilution of the observed RsVC could not be quantified. Consequently, the real difference between SCs and CAs is underestimated by the observed RsVC = 0.643 (i.e. the real RsVC is less than 0.643). Furthermore, it must be acknowledged that this misclassification could be strong in certain SCs, where the segregation effect could be seriously underestimated. Therefore, there could be nonidentified high-risk SCs. Altogether, the segregation effect suggested by an RsVC of 0.643 is valid, and the identified high-risk SCs are real high-risk populations.

In this study, calculations for cVC, sVC and RsVC were made with the assumption that SC localization has not changed from the last Hungarian census collection. Since the most recent,

**Table 3. Impact of segregation on COVID-19 vaccination coverage in high-risk segregated colonies (HRSC\*).**

| Age groups | Number of HRSC | Population in HRSC | Number of vaccinations in HRSC | Number of excess cases in HRSC | Attributable risk in HRSC |
|---|---|---|---|---|---|
| **18–59 years** | 412 | 159,433 | 52,578 | -40,091.4 | -76.3% |
| **60+ years** | 85 | 14,455 | 7,705 | -2,862.6 | -37.2% |
| **18+ years** | 437 | 208,494 | 80,065 | -47,494.5 | -59.3% |

\*standardized vaccination coverage in the settlement's segregated colony significantly lower than that in complementary part of the same settlement.

census took place in 2011, both geographic boundaries and demographic composition of SCs and CAs may have changed due to internal migration. This effect can be evaluated having the data from the next census, which will be implemented in 2022.

Taking into consideration that SCs are dominated but not exclusively inhabited by Roma, and about three quarters of the Roma population does not reside in SCs, our findings of SC vaccination inequality cannot be extrapolated to characterize Roma vaccination inequality in general.

This study was not able to collect data on many factors that can affect vaccination coverage apart from age, sex, and exemption certificate eligibility [42]. A detailed exploration on the background of SC-specific vaccination's variability needs further study. The social, cultural, economic and/or environmental differences between settlements with vaccination inequalities in SCs and those without inequality could be confounding variables in the rates of vaccination in these communities. Thus, there must be further understandings of the ethnic makeup and social determinants of health in Hungarian SCs.

## Implications

The detected inequality demonstrates the necessity of special intervention programs targeting SCs in Hungary. Clearly, the general program was not as effective in SCs. The general program considered elderly individuals, patients with chronic disease, and health care personnel as high-risk groups. It appears that the population living in SCs, comprised dominantly of Roma, is also a high-risk group. Therefore, the population living in SCs should be considered high-risk target groups, where the general vaccination program should be adapted.

Extending the high-risk group definition in Hungary could help to mitigate the impacts of the epidemic on disadvantaged, mainly Roma minority of SCs. Although, there are obvious differences between Hungarian SC inhabitants and the ethnic minorities of the United States and the United Kingdom, this reconsideration is supported by published positive experiences and elaborated methodologies in these countries where vaccination organizations and monitoring are adapted to racial/ethnic minorities [6, 43–48].

It is also urgent to expand these methods in Hungary to reduce COVID-19 vaccination inequalities. If it is successful, then the approach could be used as a template for other inequality reductions. Otherwise, if a special intervention program is not implemented and applied, the lesson by the end of the pandemic will be that the COVID-19 epidemic (and other problems) can be managed without considering ethnic inequalities and accepting the associated additional health loss.

One of the seven objectives of the European Union is to "Increase effective equal access to adequate desegregated housing and essential services" showing that segregation is among the most important characteristics of European Roma. It is highly probable that segregated colonies of Roma in other European countries face similar problems demonstrated in Hungary [34]. Moreover, our observations could help to characterize inequities occurring in similarly segregated groups around the world. For example, in the United States, Native American populations on reservations closely resemble the geographic and ethnic isolation patters exhibited by the Roma in Hungary [49]. Thus, understanding vaccination discrimination in the Roma-populated SCs parallels the in access to healthcare and vaccination disparities of minority populations in the United States [50, 51].

## Conclusions

The COVID-19 vaccination program was less effective in Hungarian SCs, populated mainly by Roma, than in the nonsegregated part of the settlements with SCs. Accordingly, the population

of SCs can be considered a distinct high-risk group with respect to vaccination coverage. This inequality was weaker but significant among elderly individuals. On the other hand, there was not detectable disparities in all settlements, demonstrating that there were effective COVID-19 vaccination protocols in Hungary, which avoided SC related geographical inequality which can be an indicator of Roma vs. non-Roma discrimination.

Our findings suggest that within the SC population, elderly individuals, patients with chronic disease, and healthcare personnel should be considered as distinct target groups with adapted vaccination programs. The required adapted methodology could be based on the benchmarking of good practice by which vaccination rate disparities were avoided in many Hungarian settlements and on the experiences reported in other countries. The need for SC-adapted methodology should be as clear as the need for a special vaccination program for healthcare personal.

## Supporting information

**S1 Checklist. Inclusivity in global research.**
(DOCX)

## Acknowledgments

This research would not have been possible without the IT support provided by Tibor Jenei (Department of Public Health and Epidemiology, Faculty of Medicine, University of Debrecen, Debrecen, Hungary), Zsófia Falusi (National Health Insurance Fund, Budapest, Hungary), and László Pál (National Health Insurance Fund, Budapest, Hungary).

## Author Contributions

**Conceptualization:** János Sándor, László Ulicska, Karolina Kósa, Róza Ádány.

**Formal analysis:** János Sándor, Ferenc Vincze, László Kőrösi.

**Investigation:** János Sándor, Ferenc Vincze, László Kőrösi.

**Methodology:** János Sándor, Ferenc Vincze, László Kőrösi.

**Project administration:** János Sándor.

**Resources:** János Sándor, Róza Ádány.

**Software:** Ferenc Vincze.

**Supervision:** János Sándor.

**Validation:** János Sándor, Ferenc Vincze, Maya Liza Shrikant.

**Writing – original draft:** János Sándor, Maya Liza Shrikant.

**Writing – review & editing:** János Sándor, Ferenc Vincze, Maya Liza Shrikant, László Kőrösi, László Ulicska, Karolina Kósa, Róza Ádány.

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
