## [Decision Letter · Decision Letter 0]

15 Nov 2021

PONE-D-21-32644COVID-19 vaccination coverage in deprived populations living in segregated colonies: a nationwide cross-sectional study in HungaryPLOS ONE

Dear Dr. Sándor,

Thank you for submitting your manuscript to PLOS ONE. After careful consideration, we feel that it has merit but does not fully meet PLOS ONE’s publication criteria as it currently stands. Therefore, we invite you to submit a revised version of the manuscript that addresses the points raised during the review process. Please address these comments one by one, particularly on the relevance of comparing Roma with other ethnic minorities outside Europe.

We look forward to receiving your revised manuscript.

Kind regards,

Joël Mossong

Academic Editor

PLOS ONE

Journal Requirements:

3. Please include a complete copy of PLOS’ questionnaire on inclusivity in global research in your revised manuscript. Our policy for research in this area aims to improve transparency in the reporting of research performed outside of researchers’ own country or community. The policy applies to researchers who have travelled to a different country to conduct research, research with Indigenous populations or their lands, and research on cultural artefacts. The questionnaire can also be requested at the journal’s discretion for any other submissions, even if these conditions are not met.  Please find more information on the policy and a link to download a blank copy of the questionnaire here: https://journals.plos.org/plosone/s/best-practices-in-research-reporting. Please upload a completed version of your questionnaire as Supporting Information when you resubmit your manuscript.

5. Thank you for stating the following in the Funding Section of your manuscript: 

"This study was carried out in the framework of the “ Routine monitoring for the health status and health care use in the Hungarian segregated colonies” program (BM/6327-3/2021, FEIF/951/2021-ITM), supported by the Deputy State Secretariat for Social Inclusion, Ministry of Interior."

"This study was carried out in the framework of the “ Routine monitoring for the health status and health care use in the Hungarian segregated colonies” program (BM/6327-3/2021, FEIF/951/2021-ITM), supported by the Deputy State Secretariat for Social Inclusion, Ministry of Interior. (https://2010-2014.kormany.hu/en/ministry-of-interior)

JS, FV, LK, KK, RÁ received grant from that program.

The funder had no role in study design, data collection and analysis, decision to publish, or preparation of the manuscript."

6. We note that you have indicated that data from this study are available upon request. PLOS only allows data to be available upon request if there are legal or ethical restrictions on sharing data publicly. For more information on unacceptable data access restrictions, please see http://journals.plos.org/plosone/s/data-availability#loc-unacceptable-data-access-restrictions. 

Reviewers' comments:

Reviewer's Responses to Questions

**Comments to the Author**

1. Is the manuscript technically sound, and do the data support the conclusions?

Reviewer #1: Yes

Reviewer #2: Yes

2. Has the statistical analysis been performed appropriately and rigorously? 

Reviewer #1: I Don't Know

Reviewer #2: Yes

3. Have the authors made all data underlying the findings in their manuscript fully available?

Reviewer #1: No

Reviewer #2: Yes

4. Is the manuscript presented in an intelligible fashion and written in standard English?

Reviewer #1: No

Reviewer #2: Yes

5. Review Comments to the Author

Reviewer #1: The authors, through a cross-sectional study, provide evidence of ethnic inequalities in vaccination coverage in Hungary. The manuscript is supported by empirical evidence collected at national level; the conclusions are based on the data presented. The authors state that restrictions will apply in the availability of data underpinning this study.

This study is a very relevant piece of research, especially in the context of multicultural societies, where disadvantaged minority ethnic groups experience growing health inequalities, which have been further deepened by the COVID-19 Pandemic.

I cannot comment on statistical analysis as my expertise is in qualitative data collection/analysis, rather than quantitative methodologies.

This manuscript requires substantial copyediting to improve readability, clarity and accuracy of the contents/statements. The uploaded pdf document contains notes with additional comments, and sentences highlighted in yellow where improvement might be necessary.

All in all it is an interesting and relevant piece of research. A thorough editing would greatly improve the quality of this manuscript.

Reviewer #2: 1. Line 62: Please elaborate further on how the similarity between CEE Roma and Black populations in the US and UK (with respect to health care use) are probable? Based on what assumptions, or research? Based on the social, environmental and other determinants of health? Or is it something else?

2. Line 227: Reported experiences from other countries about the inequality-generating characteristics of COVID-19 programs are not supported by reference.

3. Line 263: Lacks those Roma that might be without valid ID documents and/or registered address.

4. Line 272: I find the assumption of SC localisation not having changed from the last Hungarian census collection (2011) vague.

5. Line 289: SCs are to be targeted with interventions programs regardless of ethnicity, as due to the lack of data on ethnicity, we don't know the real number of the Roma population living there. Using geographical units as indicators might not be accurate.

6. Line 298: All references refer to black and/or asian communities, and the section lacks a deeper discussion on how their challenges might be similar to those of the Roma or why they are relevant.

7. Line 310: The statement that USA Native American populations on reservations closely resemble the geographic and ethnic isolation patterns exhibited by the Roma in Hungary lacks a reference, and suggests that this statement is based on subjective assumptions. It should be either supported by scientific sources and discussed in more details, or should be removed from the text.

General remarks:

Data based on health indicators of geographical units, even if these units are mainly populated by Roma might result in bias as many Roma don't live in these areas across Hungary. Thus, the research reveals data and information on the populations of the SCs and CAs, regardless of their ethnic background.

Furthermore, the research does not take non-governmental initiatives into consideration in terms of vaccination campaigns, that were carried out mainly in the most deprived and segregated areas of Hungary.

6. PLOS authors have the option to publish the peer review history of their article (what does this mean?). If published, this will include your full peer review and any attached files.

Reviewer #1: No

Reviewer #2: **Yes: **Bernadett Varga, M.PH

---

## [Author Response · Author response to Decision Letter 0]

15 Dec 2021

Dear Editor and Reviewers,

Thank you very much for the careful review of our manuscript. Please find enclosed the revised version of the manuscript “COVID-19 vaccination coverage in deprived populations living in segregated colonies: a nationwide cross-sectional study in Hungary” by János Sándor, et al.

Each comment and suggestion has been considered. The corresponding changes and refinements made in the revised paper are summarized in our response after considering each of your suggestion. Answers along with the modifications we made are summarized below (comments/questions of Yours are in capitals).

Sincerely yours, Janos Sandor (on behalf of the authors)

Answers/reflections to the comments of the Editor:

1. The title page and the main text have been corrected according to the PLOS ONE style templates.

2. Reference list has been updated according to the corrections. New references were added (see answers to the specific comments to the reviewers), and the numbering of references has been updated.

3. We completed the PLOS’ questionnaire on inclusivity in global research. It has been uploaded as Supporting Information (S1 Checklist.docx). The suggested subsection has been added to the Methods section: “Additional information regarding the ethical, cultural, and scientific considerations specific to inclusivity in global research is included in the Supporting Information (S1 Checklist)”

4. The correct information on funding is the following:

"This study was carried out in the framework of the “ Routine monitoring for the health status and health care use in the Hungarian segregated colonies” program (BM/6327-3/2021, FEIF/951/2021-ITM), supported by the Deputy State Secretariat for Social Inclusion, Ministry of Interior. (https://2010-2014.kormany.hu/en/ministry-of-interior)

JS, FV, LK, KK, RÁ received grant from that program.

The funder had no role in study design, data collection and analysis, decision to publish, or preparation of the manuscript."

Please, modify the statement in the on-line submission system.

5. The Funding section of our manuscript has been deleted.

6. Because, we would like to respect the PLOS Data policy, the database of the investigation with the segregated colony and complementary area specific observed and expected number of completed vaccinations had been archived in the figshare repository.

This file can be downloaded without any restriction. All the indicators reported in our manuscript can be produced from this dataset. The name of settlements have been replaced with serial numbers in the dataset, because the identification of a settlement is not allowed: the Hungarian law on statistical data reporting (2016. évi CLV. törvény a hivatalos statisztikáról; Act on Official Statistics 2016) and related decrees prohibit the report of statistical data with less than or equal to 5 number of observed cases, and there are many observed number of cases meeting this criterion in the dataset. (The ’data.xlsx’ file is available searching for ’covid-19 vaccination’ and applying the following filters: Licence CC BY 4.0; Item Type dataset; Source figshare.com; Category Preventive Medicine.)

7. Files with figures had been uploaded to the PACE. The original figures had been revised according to this list:

Figure1.tiff: Resolution is changed to 300 PPI. Dimensions are adjusted to 2.63in W x 2.63in H TIFF file is converted to a valid TIF file.

Figure2.tiff: Resolution is changed to 300 PPI. TIFF file is converted to a valid TIF file.

Figure3a.tif: Compression is set to LZW. Resolution is changed to 300 PPI Dimensions are adjusted to 7.5in W x 3.91in H TIF file is converted to a valid TIF file.

Figure3b.tif: Compression is set to LZW. Resolution is changed to 300 PPI Dimensions are adjusted to 7.5in W x 3.91in H TIF file is converted to a valid TIF file.

Figure3c.tif: Compression is set to LZW. Resolution is changed to 300 PPI Dimensions are adjusted to 7.5in W x 3.91in H TIF file is converted to a valid TIF file.

The original files have been replaced with the PACE-revised versions in the on-line submission system.

Answers/reflections to the comments of Reviewer-1:

1.

THE AUTHORS, THROUGH A CROSS-SECTIONAL STUDY, PROVIDE EVIDENCE OF ETHNIC INEQUALITIES IN VACCINATION COVERAGE IN HUNGARY. THE MANUSCRIPT IS SUPPORTED BY EMPIRICAL EVIDENCE COLLECTED AT NATIONAL LEVEL; THE CONCLUSIONS ARE BASED ON THE DATA PRESENTED. THE AUTHORS STATE THAT RESTRICTIONS WILL APPLY IN THE AVAILABILITY OF DATA UNDERPINNING THIS STUDY.

Because, we would like to respect the PLOS Data policy, the database of the investigation with the segregated colony and complementary area specific observed and expected number of completed vaccinations had been archived in a public available site of the University of Debrecen:

https://nepegeszseg.unideb.hu/data/

This file can be downloaded without any restriction. All the indicators reported in our manuscript can be produced from this dataset. The name of settlements have been replaced with serial numbers in the dataset, because the identification of a settlement is not allowed: the Hungarian law on statistical data reporting (2016. évi CLV. törvény a hivatalos statisztikáról; Act on Official Statistics 2016) and related decrees prohibit the report of statistical data with less than or equal to 5 number of observed cases, and there are many observed number of cases meeting this criterion in the dataset.

2.

THIS STUDY IS A VERY RELEVANT PIECE OF RESEARCH, ESPECIALLY IN THE CONTEXT OF MULTICULTURAL SOCIETIES, WHERE DISADVANTAGED MINORITY ETHNIC GROUPS EXPERIENCE GROWING HEALTH INEQUALITIES, WHICH HAVE BEEN FURTHER DEEPENED BY THE COVID-19 PANDEMIC.

Thank you very much for this evaluation!

3.

I CANNOT COMMENT ON STATISTICAL ANALYSIS AS MY EXPERTISE IS IN QUALITATIVE DATA COLLECTION/ANALYSIS, RATHER THAN QUANTITATIVE METHODOLOGIES.

---

4.

THIS MANUSCRIPT REQUIRES SUBSTANTIAL COPYEDITING TO IMPROVE READABILITY, CLARITY AND ACCURACY OF THE CONTENTS/STATEMENTS. THE UPLOADED PDF DOCUMENT CONTAINS NOTES WITH ADDITIONAL COMMENTS, AND SENTENCES HIGHLIGHTED IN YELLOW WHERE IMPROVEMENT MIGHT BE NECESSARY.

The text has been edited by the American Journal Expert (certificate uploaded to the on-line submission system).The highlighted texts with comments are reflected point by point as below:

Line 22:

THIS SENTENCE SHOULD BE REFORMULATED. “WHETHER FAIR COVID-19 VACCINATION COVERAGE HAS BEEN ACHIEVED IN HUNGARY IS ACCOMPANIED WITH FAIR EFFECTIVENESS IN SCS, HAS NOT YET BEEN STUDIED.”

Corrected version:

It has not been studied yet, whether fair COVID-19 vaccination coverage achieved in Hungary is accompanied with fair effectiveness in SCs.

Line 32:

THIS SENTENCE SHOULD BE REFORMULATED. “HUNGARIAN SCS POPULATED MAINLY BY ROMA ARE A DISTINCT HIGH-RISK GROUP WITH RESPECT TO COVID-19 VACCINATION.”

Thank you for this suggestion!

Corrected version:

Hungarian citizens living in SCs, mainly of Roma ethnicity, are a distinct high-risk group.

Line 38:

IT WOULD BE ADVISABLE TO SPELL OUT THE MEANING OF THIS ACRONYM (SARS-COV-2) THE FIRST TIME IT IS MENTIONED IN THE MANUSCRIPT.

Full name of the virus has been inserted.

Corrected version:

Severe Acute Respiratory Syndrome Coronavirus-2 (SARS-CoV-2)

Line 43:

I THINK THE WORD “AND” IS NOT NEEDED HERE; IT SEEMS TO MISLEAD THE CORRECT MEANING OF THIS SENTENCE.

The “and” has been deleted. Thank you for this suggestion!

Corrected version:

“When and where COVID-19 vaccines became available, the vaccination programs’ target groups were determined by taking into consideration the susceptibility of individuals to lethal complications …”

Line 52:

THIS REMARK IS VERY IMPORTANT, HOWEVER, THE WAY IT IS WRITTEN IS MORE SUITABLE AS A RECOMMENDATION FROM STUDY FINDINGS. THIS SENTENCE AND THE NEXT COULD BE MERGED AS FOLLOWS: "THIS EVIDENCE SUPPORTS THE CASE FOR STRICT MONITORING OF RACIAL/ETHNIC INEQUALITIES IN VACCINATION UPTAKE/COVERAGE, WHICH IS CONFIRMED BY PUBLISHED REPORTS ON SUCCESSFUL INTERVENTIONS IN THIS REGARD."

Thanks for the suggested reformulation!

Corrected version:

“This evidence supports the case for strict monitoring of racial/ethnic inequalities in vaccination uptake/coverage, which is confirmed by published reports on successful interventions in this regard.”

Line 53:

REFERENCES SHOULD BE PLACED BEFORE FULL STOP THROUGHOUT THE MANUSCRIPT.

Correction:

All the references have been placed before full stop.

Line 55:

CULTURALLY APPROPRIATE?

The suggested completion has been accepted.

Corrected version:

“… that require minority-adapted, culturally appropriate approaches …”

Line 57:

THIS SENTENCE COULD END LIKE THIS "....VACCINE-PREVENTABLE COVID-19 INFECTIONS"

Corrected version:

“… to diminish the consequences of the vaccine-preventable COVID-19 infections in the forthcoming wave(s) of the pandemic.”

Line 59:

"....WHO ARE THE LARGEST ETHNIC MINORITY IN EUROPE"?

Corrected version:

“… home to 10-12 million Roma (many of them in segregated colonies, SC) who are the largest ethnic minority in Europe.”

Line 60:

IN HUNGARY OR IN CEE?

There is specific monitoring neither in CEE nor in Hungary.

Corrected version:

“There are no socioeconomic status (SES) or ethnic minority-specific monitoring programs either for COVID-19 health impacts or COVID-19 vaccination effectiveness in CEE countries”.

(see further modification of this sentence by the Line62 comments)

Line 62:

IT MAY BE WORTH MENTIONING HOW IN OTHER EUROPEAN COUNTRIES NGOS SUPPORTING THE RIGHTS TO HEALTH OF ROMA HAVE BEEN ADVOCATING FOR 'ETHNIC IDENTIFIER' IN COVID-19 HEALTH IMPACT. THIS MAY STRENGTHEN THE ARGUMENTS PUT FORWARD IN THIS MANUSCRIPT.

SEE HERE: HTTPS://WWW.PAVEEPOINT.IE/STAY-SAFE-FROM-CORONAVIRUS-COVID-19/

AND ALSO RECOMENDATION N.10 IN THE FOLLOWING REPORT: HTTPS://WWW.PAVEEPOINT.IE/WP-CONTENT/UPLOADS/2015/04/VACCINE-IMPLEMENTATION.PDF

Thank you very much for these suggestions! It really adds to the paper! One of the proposed references has been inserted. (Fay R, Kavanagh L, Amin N. COVID-19 and Irish travellers: Interim responses, reflections and recommendations. Available from: https://www.paveepoint.ie/wp-content/uploads/2015/04/COVIDREPORTWEB.pdf Accessed at 24/11/2021)

Corrected version:

“In spite of some positive European experiences on Roma specific monitoring [REF], there are no socioeconomic status (SES) or ethnic minority-specific monitoring programs either for COVID-19 health impacts or COVID-19 vaccination effectiveness in CEE countries.”

Line 64:

The same NGOs have also published evidence of targeted COVID-19 pandemic mitigation interventions for Travellers and Roma in Ireland which may be worthwhile mentioning here as good practice in other European countries: https://journals.sagepub.com/doi/full/10.1177/1757975921994075

Thank you very much for this suggestion! Because this paper is about Roma specific interventions, citation of this paper is really important. But, the sentence “To date, there are published reports on successful interventions aimed at diminishing racial/ethnic disparities in vaccination coverage.”, therefore this reference has been added to the references for this sentence.

Villani J, Daly P, Fay R, Kavanagh L, McDonagh S, Amin N. A community-health partnership response to mitigate the impact of the COVID-19 pandemic on Travellers and Roma in Ireland. Glob Health Promot. 2021; 28(2):46-55. doi: 10.1177/1757975921994075

Correction:

Reference has been added.

Line 75:

THIS SENTENCE NEEDS TO BE REFORMULATED. “THIS ESTABLISHES THE PRODUCTION OF SC-SPECIFIC INDICATORS BY WHICH THE NEEDS’ EXPLORATION AND THE ELABORATION OF INTERVENTIONS, AS WELL AS AN ASSESSMENT OF THE INTERVENTIONS’ EFFECTIVENESS CAN BE CARRIED OUT.”

Corrected version:

“It establishes the production of indicators by which the needs can be explored and interventions can be elaborated for SCs, as well as the effectiveness of implemented SC-specific programs can be assessed.

Line 84:

ANY REFERENCE AVAILABLE TO SUPPORT THIS STATEMENT?

The ECDC regularly updates a downloadable dataset of the country specific indicators of vaccination.

Corrected version:

A new reference has been added as:

European Centre for Disease Prevention and Control: Download COVID-19 datasets; https://www.ecdc.europa.eu/en/covid-19/data Accessed at 30/09/2021

Line 87:

IT WAS NEVER EXPLAINED BEFORE THAT THERE IS A NONSEGREGATED PART OF THE SETTLEMENTS. PERHAPS THIS COULD BE EXPLAINED FOR READERS WHO ARE NOT FAMILIAR WITH THE ROMA SETTLEMENTS IN HUNGARY.

Corrected version:

“… in relation to the nonsegregated part of the same settlements …”

Line 90:

WHILE THE TITLE OF THE MANUSCRIPT REFLECTS THE STUDY DESIGN (I.E. CROSS-SECTIONAL STUDY), THIS IS NOT MENTIONED EXPLICITLY IN THE MANUSCRIPT. THIS COULD BE SPECIFIED IN THE TEXT, ALONG WITH A REFERENCE OF SIMILAR STUDIES USING THE SAME DESIGN. THIS WILL STRENGTHEN THE METHODOLOGICAL UNDERPINNINGS OF THE STUDY.

Thank you for this suggestion! The design has been inserted into the main text, into the first sentence of the Setting section.

Corrected version:

This cross-sectional investigation encompassed the whole country.

Line 99:

THIS PARAGRAPH REQUIRES EDITING. THE DEFINITIONS OF SC AND CA ARE VERY IMPORTANT FOR THE PURPOSE OF THIS PAPER, AND THEY SHOULD BE DEFINED MORE CLEARLY. THIS WILL HELP READERS WHO ARE NOT FAMILIAR WITH THE HUNGARIAN ENVIRONMENT. FOR EXAMPLE, LATER ON IT IS EXPLAINED THAT IN A 'SETTLEMENT' THERE COULD BE SEVERAL SCS. PLEASE DEFINE WHAT A 'SETTLEMENT' IS.

Corrected version:

A governmental decree defines SCs as within settlement (within towns and within villages) clustering of residents 18-59 years old with not higher than primary level education and a lack of work-related income. The Hungarian Central Statistical Office determines the cluster or clusters as SC(s) and the complementary areas (CA) of the same settlements not belonging to any SC for all Hungarian settlements.

Line 123:

Nationality?

Not. But, this misleading sentence has been modified.

Corrected version:

Age, sex, and eligibility for exemption certificate-specific reference vaccination ratios for the whole population of Hungary were calculated and used in indirect standardization.

Line 130:

DEFINITION OF 'SETTLEMENT' IS REQUIRED.

The term settlement as the aggregated concept for villages and towns has been explained before (see the answer to the Line 99 comment)

This sentence has not been modified.

Line 149

THIS SENTENCE CAN BE MERGED WITH THE PREVIOUS ONE: "..."

Corrected version:

A total of 4,987,661 persons (60.7% of the Hungarian adult population) live in settlements with SCs, and 276,879 (5.6%) of them live in SCs.

Line 226:

REFERENCE?

New references had been added. References cited in the Introduction (references 17-23) have been cited here again.

Line 236:

THE INTENT OF THE AUTHORS TO DEMONSTRATE INEQUALITIES IN VACCINATION COVERAGE IS CLEAR HERE. HOWEVER, THIS PARAGRAPH REQUIRES EDITING TO MAKE THE SENTENCES CLEARER AND ITS CONTENT STRONGER AND MORE READABLE.

Corrected version:

By the time of our investigation, a total of 165,988 adults (9.24% of the sensitive population) were from the SCs and 1,629,216 from CAs. If the vaccination coverage in SCs were equal to that in CAs, then there would be only 95,758 susceptible inhabitants in SCs. It would be reflected in 3.91% decrease of the number of susceptible adults in the studied population, and in the 5.55% of the susceptible inhabitants would be from SCs. Consequently, although, the pandemic related future health loss could be prevented basically by the elevation of the vaccination coverage in the CAs, a not negligible risk reduction could be achieved by the enforced/adapted vaccination project in the SCs.

Line 293:

'THE POPULATION LIVING IN SCS SHOULD BE CONSIDERED HIGH RISK TARGET GROUPS'

Thank you for this suggestion!

Corrected version:

“Therefore, the population living in SCs should be considered high risk target groups, where the general vaccination program should be adapted.”

Line 296:

'...ON DISADVANTAGE ETHNIC MINORITIES'?

Corrected version:

“… to mitigate the impacts of the epidemic on disadvantaged ethnic minorities.”

Line 310:

IT WOULD BE RECOMMENDED TO INTRODUCE THE EXAMPLE OF NATIVE AMERICANS, AND RELATED REFERENCES, IN THE INTRODUCTION ALONG WITH BLACKS IN US AND UK (LINES 62-64), AS THIS IS THE FIRST TIME WHERE NATIVE AMERICANS ARE MENTIONED IN THE MANUSCRIPT, AT THE VERY END OF THE MANUSCRIPT. GENERALLY THE DISCUSSION DOES NOT INTRODUCE NEW CONCEPTS OR TOPICS.

Mention of Native Americans has been added to the sentence suggested with a new reference.

Cromer KJ, Wofford L, Wyant DK. Barriers to Healthcare Access Facing American Indian and Alaska Natives in Rural America. J Community Health Nurs. 2019;36(4):165-187. doi: 10.1080/07370016.2019.1665320

Corrected version:

“Although it seems probable that the CEE Roma are similar to the Black populations in the US and UK as well as to the Native Americans in the US with respect to health care use [REF], …”

Line 315:

FINALLY, THE MANUSCRIPT REQUIRES SIGNIFICANT PROOFREADING AND COPYEDITING AS SEVERAL PARAGRAPHS THROUGHOUT THE TEXT ARE NOT CLEAR. THE AUTHORS COULD SEEK EDITORIAL HELP BEFORE SUBMITTING A REVISION.

Thank you very much for the careful reviewing! The criticized texts have been modified (as it is reported above). We hope that these reflections are acceptable. (The text has been edited by the American Journal Expert -certificate uploaded to the on-line submission system.)

5.

ALL IN ALL IT IS AN INTERESTING AND RELEVANT PIECE OF RESEARCH. A THOROUGH EDITING WOULD GREATLY IMPROVE THE QUALITY OF THIS MANUSCRIPT.

Thank you for this general evaluation! The required editing has been completed as detailed above.

Answers/reflections to the comments of Reviewer-2:

Line 62:

PLEASE ELABORATE FURTHER ON HOW THE SIMILARITY BETWEEN CEE ROMA AND BLACK POPULATIONS IN THE US AND UK (WITH RESPECT TO HEALTH CARE USE) ARE PROBABLE? BASED ON WHAT ASSUMPTIONS, OR RESEARCH? BASED ON THE SOCIAL, ENVIRONMENTAL AND OTHER DETERMINANTS OF HEALTH? OR IS IT SOMETHING ELSE?

As it is stated in the sentence, the restricted health care availability is the common characteristic of the mentioned minorities. (The paper is about the restricted access to a special health care service – COVID-19 vaccination. The paper is not about the mechanisms behind the inequality.) To support this statement two references had been added to the text.

Williams DR, Rucker TD. Understanding and Addressing Racial Disparities in Health Care. Health Care Financ Rev. 2000; 21(4):75–90.

Fiscella K, Sanders MR. Racial and Ethnic Disparities in the Quality of Health Care. Annu Rev Public Health. 2016;37:375-94. doi: 10.1146/annurev-publhealth-032315-021439

Correction:

New references added.

Line 227:

REPORTED EXPERIENCES FROM OTHER COUNTRIES ABOUT THE INEQUALITY-GENERATING CHARACTERISTICS OF COVID-19 PROGRAMS ARE NOT SUPPORTED BY REFERENCE.

The same was requested by the Reviewer-1 also. References cited in the Introduction (references 17-23) have been cited here again. These papers are about the observed inequalities.

Correction:

New references added.

Line 263:

LACKS THOSE ROMA THAT MIGHT BE WITHOUT VALID ID DOCUMENTS AND/OR REGISTERED ADDRESS.

Because the study base of this investigation was adults with registered residential place (it is accompanied with valid ID) in settlements with at least one segregated colony, people belong to other part of the Hungarian population, even being at high-risk in the respect of omitting COVID-19 vaccination, were not scrutinized.

No correction in the text.

Line 272:

I FIND THE ASSUMPTION OF SC LOCALISATION NOT HAVING CHANGED FROM THE LAST HUNGARIAN CENSUS COLLECTION (2011) VAGUE.

The limitation caused by the not updated segregation boundaries is explicitly acknowledged in the paragraph (4th paragraph in the Strengths and limitations section).

No correction in the text.

Line 289:

SCS ARE TO BE TARGETED WITH INTERVENTIONS PROGRAMS REGARDLESS OF ETHNICITY, AS DUE TO THE LACK OF DATA ON ETHNICITY, WE DON'T KNOW THE REAL NUMBER OF THE ROMA POPULATION LIVING THERE. USING GEOGRAPHICAL UNITS AS INDICATORS MIGHT NOT BE ACCURATE.

This limitation is also acknowledged in the paragraph of Line 277-280: “Taking into consideration that SCs are dominated but not exclusively inhabited by Roma, and about three quarters of the Roma population does not reside in SCs, our findings of SC vaccination inequality cannot be extrapolated to characterize Roma vaccination inequality in general.” We did not (over-)interpret our results as Roma specific findings.

No correction in the text.

Line 298:

ALL REFERENCES REFER TO BLACK AND/OR ASIAN COMMUNITIES, AND THE SECTION LACKS A DEEPER DISCUSSION ON HOW THEIR CHALLENGES MIGHT BE SIMILAR TO THOSE OF THE ROMA OR WHY THEY ARE RELEVANT.

It was not among the objectives of our investigation to explore the background of SC-related inequality in COVID-19 vaccination in Hungary. The reliable description of this inequality was aimed to be demonstrated. The purpose with reference to similar inequality in other countries was to demonstrate that the Hungarian finding is not exceptional according to the international experiences.

No correction in the text.

Line 310:

THE STATEMENT THAT USA NATIVE AMERICAN POPULATIONS ON RESERVATIONS CLOSELY RESEMBLE THE GEOGRAPHIC AND ETHNIC ISOLATION PATTERNS EXHIBITED BY THE ROMA IN HUNGARY LACKS A REFERENCE, AND SUGGESTS THAT THIS STATEMENT IS BASED ON SUBJECTIVE ASSUMPTIONS. IT SHOULD BE EITHER SUPPORTED BY SCIENTIFIC SOURCES AND DISCUSSED IN MORE DETAILS, OR SHOULD BE REMOVED FROM THE TEXT.

Thank you very much for this comment! A comprehensive health status evaluation of Native American community is presented in this report: Indian Health Service. Trends in Indian health - 2014 edition.

Available from https://www.ihs.gov/sites/dps/themes/responsive2017/display_objects/documents/Trends2014Book508.pdf Accessed at 24/11/2021

Correction:

New reference has been added.

General remarks 1:

DATA BASED ON HEALTH INDICATORS OF GEOGRAPHICAL UNITS, EVEN IF THESE UNITS ARE MAINLY POPULATED BY ROMA MIGHT RESULT IN BIAS AS MANY ROMA DON'T LIVE IN THESE AREAS ACROSS HUNGARY. THUS, THE RESEARCH REVEALS DATA AND INFORMATION ON THE POPULATIONS OF THE SCS AND CAS, REGARDLESS OF THEIR ETHNIC BACKGROUND.

This remark is true. But, it was not the aim of our investigation to describe the vaccination among Roma compared with the vaccination among non-Roma in general. To demonstrate that we did not over-interpret the presented results, we explicitly stated it in the text (5th paragraph in Strengths and limitations section) that: “Taking into consideration that SCs are dominated but not exclusively inhabited by Roma, and about three quarters of the Roma population does not reside in SCs, our findings of SC vaccination inequality cannot be extrapolated to characterize Roma vaccination inequality in general.”

No correction in the text.

General remarks 2:

FURTHERMORE, THE RESEARCH DOES NOT TAKE NON-GOVERNMENTAL INITIATIVES INTO CONSIDERATION IN TERMS OF VACCINATION CAMPAIGNS, THAT WERE CARRIED OUT MAINLY IN THE MOST DEPRIVED AND SEGREGATED AREAS OF HUNGARY.

This paper is not about the evaluation of the background of the vaccination inequality. It is a descriptive paper about the demonstration of within settlement inequality. The last paragraph of the Strengths and limitations section is about the future research needs: “This study was not able to collect data on many factors that can affect vaccination coverage apart from age, sex, and exemption certificate eligibility [36]. A detailed exploration on the background of SC-specific vaccination’s variability needs further study. The social, cultural, economic and/or environmental differences between settlements with vaccination inequalities in SCs and those without inequality could be confounding variables in the rates of vaccination in these communities. Thus, there must be further understandings of the ethnic makeup and social determinants of health in Hungarian SCs.”

No correction in the text.

---

## [Decision Letter · Decision Letter 1]

31 Jan 2022

PONE-D-21-32644R1COVID-19 vaccination coverage in deprived populations living in segregated colonies: a nationwide cross-sectional study in HungaryPLOS ONE

Dear Dr. Sándor,

Thank you for submitting your manuscript to PLOS ONE. After careful consideration, we feel that it has merit but does not fully meet PLOS ONE’s publication criteria as it currently stands. Therefore, we invite you to submit a revised version of the manuscript that addresses the points raised during the review process.

Please address the remaining minor comments.==============================

We look forward to receiving your revised manuscript.

Kind regards,

Joël Mossong

Academic Editor

PLOS ONE

Journal Requirements:

Reviewers' comments:

Reviewer's Responses to Questions

**Comments to the Author**

1. If the authors have adequately addressed your comments raised in a previous round of review and you feel that this manuscript is now acceptable for publication, you may indicate that here to bypass the “Comments to the Author” section, enter your conflict of interest statement in the “Confidential to Editor” section, and submit your "Accept" recommendation.

Reviewer #1: All comments have been addressed

Reviewer #2: All comments have been addressed

2. Is the manuscript technically sound, and do the data support the conclusions?

Reviewer #1: Yes

Reviewer #2: Yes

3. Has the statistical analysis been performed appropriately and rigorously? 

Reviewer #1: I Don't Know

Reviewer #2: I Don't Know

4. Have the authors made all data underlying the findings in their manuscript fully available?

Reviewer #1: Yes

Reviewer #2: Yes

5. Is the manuscript presented in an intelligible fashion and written in standard English?

Reviewer #1: Yes

Reviewer #2: Yes

6. Review Comments to the Author

Reviewer #1: Dear Authors,

Thank you for submitting your revised manuscript. All my comments have been properly addressed. The readability and quality of the manuscript has improved substantially; thank you for accepting the advice to avail of professional editorial support.

As stated in my first review I cannot comment on the statistical analysis as my expertise is not quantitative methods. I trust the editors and reviewer 2 have evaluated the quality and robustness of statistical analysis.

I acknowledge that authors have made the data used for this manuscript publicly available as per PLOS ONE requirements.

Best wishes.

Reviewer #2: Dear Authors, thank you for your feedback in response to my comments, I appreciate it. However, there are elements, which in my opinion still need to be revised in order to publish this very important research.

In response to my comment for Line 263 your feedback states "Because the study base of this investigation was adults with registered residential place (it is accompanied with valid ID) in settlements with at least one segregated colony, people belong to other part of the Hungarian population, even being at high-rist in the respect of omitting COVID-19 vaccination, were not scrutinized.", and

in response to my comment for Line 289 your feedback states "We did not (over-)interpret our results as Roma specific findings".

However, your abstract reads as follows: "The largest ethnic minority in Hungary is Roma, comprising 8.7% of the population. Their healthcare use is restricted in many respects. Hungarian citizens living in SCs, mainly of Roma ethnicity (...)."

There is a discrepancy between the first lines of the abstract and the rest of the text of the research. Although it is further elaborated within the text later, the abstract does not support the above statements. Please rephrase your abstract accordingly, as the text currently suggests that this research focuses on the Roma ethnic minority (living in SCs and CAs).

Furthermore, with regards to Line 298 no reference or scientific evidence has been added to support the statement that Roma and Black/Asian communities face the same challenges and obstacles in terms of their access to health services. In my opinion this part should be either extended, or rephrased.

7. PLOS authors have the option to publish the peer review history of their article (what does this mean?). If published, this will include your full peer review and any attached files.

Reviewer #1: No

Reviewer #2: **Yes: **Bernadett Varga, M.PH

---

## [Author Response · Author response to Decision Letter 1]

2 Feb 2022

Dear Editor and Reviewer,

Thank you very much for the careful second review of our manuscript. Please find enclosed the revised version of the manuscript “COVID-19 vaccination coverage in deprived populations living in segregated colonies: a nationwide cross-sectional study in Hungary” by János Sándor, et al.

Both comments have been considered. The corresponding changes made in the revised paper are summarized in our response after considering each of your suggestion. Answers along with the modifications we made are summarized below (comments/questions of Yours are in capitals).

Sincerely yours, Janos Sandor (on behalf of the authors)

Answers/reflections to the comments of the Editor:

1. 

THERE IS A DISCREPANCY BETWEEN THE FIRST LINES OF THE ABSTRACT AND THE REST OF THE TEXT OF THE RESEARCH. ALTHOUGH IT IS FURTHER ELABORATED WITHIN THE TEXT LATER, THE ABSTRACT DOES NOT SUPPORT THE ABOVE STATEMENTS. PLEASE REPHRASE YOUR ABSTRACT ACCORDINGLY, AS THE TEXT CURRENTLY SUGGESTS THAT THIS RESEARCH FOCUSES ON THE ROMA ETHNIC MINORITY (LIVING IN SCS AND CAS).

Thank you for this comment. The misleading sentences have been modified accordingly.

Original text:

“The largest ethnic minority in Hungary is Roma, comprising 8.7% of the population. A quarter of them live in segregated colonies (SCs). Their health care use is restricted in many respects.”

Corrected version:

“The segregated colonies (SCs) in Hungary are populated mainly but not exclusively by Roma. Their health care use is restricted in many respects.”

2.

FURTHERMORE, WITH REGARDS TO LINE 298 NO REFERENCE OR SCIENTIFIC EVIDENCE HAS BEEN ADDED TO SUPPORT THE STATEMENT THAT ROMA AND BLACK/ASIAN COMMUNITIES FACE THE SAME CHALLENGES AND OBSTACLES IN TERMS OF THEIR ACCESS TO HEALTH SERVICES. IN MY OPINION THIS PART SHOULD BE EITHER EXTENDED, OR REPHRASED.

As it is acknowledged in the paper our investigation could not investigate the background of SC-related inequality in COVID-19 vaccination in Hungary. The demonstration of this inequality was aimed.

We referred the successful experiences with monitoring based interventions in the field of racial inequalities’ management in order to demonstrate that the basic discipline of public health is working in practice.

We could not undertake a deeper analysis of similarities and differences between racial minorities in the USA/UK and Hungarian segregated colonies populated mainly by Roma.

To clarify this intention, the paragraph has been rephrased.

Original text:

“Extending the high-risk group definition could help to mitigate the impacts of the epidemic on disadvantaged ethnic minorities. This reconsideration is supported by published positive experiences and elaborated methodologies in countries where vaccination organizations and monitoring are adapted to racial/ethnic minorities [6,43-48].”

Corrected version:

“Extending the high-risk group definition in Hungary could help to mitigate the impacts of the epidemic on disadvantaged, mainly Roma minority of SCs. Although, there are obvious differences between Hungarian SC inhabitants and the ethnic minorities of the United States and the United Kingdom, this reconsideration is supported by published positive experiences and elaborated methodologies in countries where vaccination organizations and monitoring are adapted to racial/ethnic minorities [6,43-48].”

---

## [Editor Report · Decision Letter 2]

9 Feb 2022

COVID-19 vaccination coverage in deprived populations living in segregated colonies: a nationwide cross-sectional study in Hungary

PONE-D-21-32644R2

Dear Dr. Sándor,

We’re pleased to inform you that your manuscript has been judged scientifically suitable for publication and will be formally accepted for publication once it meets all outstanding technical requirements.

Kind regards,

Joël Mossong

Academic Editor

PLOS ONE
---

## [Editor Report · Acceptance letter]

18 Feb 2022

PONE-D-21-32644R2 

COVID-19 vaccination coverage in deprived populations living in segregated colonies: a nationwide cross-sectional study in Hungary 

Dear Dr. Sándor:

I'm pleased to inform you that your manuscript has been deemed suitable for publication in PLOS ONE. Congratulations! Your manuscript is now with our production department. 

Kind regards, 

on behalf of

Dr. Joël Mossong 

Academic Editor

PLOS ONE